# Hydrological/Hydraulic Modeling-Based Thresholding of Multi SAR Remote Sensing Data for Flood Monitoring in Regions of the Vietnamese Lower Mekong River Basin

**Nguyen Hong Quang** [1,*], **Vu Anh Tuan** [1,*], **Le Thi Thu Hang** [1], **Nguyen Manh Hung** [1],
**Doan Thi The** [1], **Dinh Thi Dieu** [1], **Ngo Duc Anh** [1] and **Christopher R. Hackney** [2,*]

[1] Vietnam National Space Center (VNSC), Vietnam Academy of Science and Technology (VAST), Nr. 18 Hoang Quoc Viet, Hanoi 100000, Vietnam; ltthang@vnsc.org.vn (L.T.T.H.); nmhung@vnsc.org.vn (N.M.H.); dtthe@vnsc.org.vn (D.T.T.); dtdieu@vnsc.org.vn (D.T.D.); ndanh@vnsc.org.vn (N.D.A.)

[2] Energy and Environment Institute, University of Hull, Hull HU6 7RX, UK

\* Correspondence: nhquang@vnsc.org.vn (N.H.Q.); vatuan@vnsc.org.vn (V.A.T.); C.Hackney@hull.ac.uk (C.R.H.); Tel.: +84-968-844-250 (N.H.Q.)

**Abstract:** Synthetic Aperture Radar (SAR) remote sensing data can be used as an effective alternative to detect surface water and provide useful information regarding operational flood monitoring, in particular for the improvement of rapid flood assessments. However, this application frequently requires standard and simple, yet robust, algorithms. Although thresholding approaches meet these requirements, limitations such as data inequality over large spatial regions and challenges in estimating optimal threshold values remain. Here, we propose a new method for SAR water extraction named Hammock Swing Thresholding (HST). We applied this HST approach to four SAR remote sensing datasets, namely, Sentinel-1, ALOS-2, TerraSAR-X, and RadarSAT-2 for flood inundation mapping for a case study focusing on the Tam Nong district in the Vietnam Mekong delta. A 2D calibrated Hydrologic Engineering Centers River Analysis System (HEC-RAS) model was coupled with the HST outputs in order to estimate the optimal thresholds (OTs) where the SAR-based water masks fitted best with HEC-RAS's inundation patterns. Our results showed that water levels extracted from Sentinel-1 data best agreed with the HEC-RAS water extent (88.3%), following by ALOS-2 (85.9%), TerraSAR-X (77.2%). and RadarSAT-2 (72%) at OTs of $-15$, 68, 21, and 35 decibel (dB), respectively. Generated flood maps indicated changes in the flood extent of the flooding seasons from 2010 and 2014–2016 with variations in spatial extent appearing greater in the TerraSAR-X and RadarSAT-2 higher resolution maps. We recommend the use of OTs in applications of flood monitoring using SAR remote sensing data, such as for an open data cube (ODC).

**Keywords:** flood mapping; hydrological/hydraulic-based thresholding; polarization effects; local incidence angle assessment; Vietnam lower Mekong basin

## 1. Introduction

Synthetic Aperture Radar (SAR) remote sensing data can be used as an effective alternative to detect surface water [1–3] and provide useful information for flood monitoring [4,5] due to its large temporal scale (e.g., decadal periods) [6]. In addition, SAR data can penetrate clouds and atmospheric interference that often appear during periods of flooding and can be used at night-time as it does not rely on spectral reflectance from the Earth's surface [7]. Furthermore, in recent years, SAR datasets have increased in their spatial and temporal resolutions as well as their data availability [8].

However, the main reason for the popular use of SAR images for flood studies is that water surfaces are able to be classified robustly due to the lower backscatter returned to the SAR sensors compared to dry areas [8].

In flood studies, which integrate SAR images and hydrological/hydraulic models, SAR data are commonly used to calibrate and validate hydrological and hydraulic models [4,9–11]. However, fewer studies have inverted this process and used the results of hydrological/hydraulic models to calibrate remote sensing-based flood extractions. Hydrological and hydraulic modeling commonly require extensive data inputs [12,13] based around four key groupings, namely, topographic, hydro-meteorological, soil, and land cover data.

There has been increasing attention paid to the large transboundary Mekong river basin due to various issues related to water management and flood control [14,15]. The Vietnamese Mekong delta (VMD) occupies 12% of Vietnam's total area and is home to approximately 18 million people [16]. Their livelihoods are reliant on rice production and aquaculture (more than 50% of Vietnam's total annual outputs come from the VMD) [17]. However, the largest risks to Vietnam's national food security are the impacts of salinization, droughts, and floods. Rising sea levels and increased catchment-wide water regulations (i.e., hydropower reservoirs and dams) in the upper regions of the Mekong Basin are contributors to these problems [17,18]; hence, to better understand changes in flooding and flood risk, improved representations of flood processes in this area are required.

Hydrological and hydraulic models are valuable tools in flood risk management and planning. By coupling these models with remote sensing-based water extent extractions, it is possible to improve model accuracy, mitigate data scarcity, and increase cost-effectiveness [6,12,19]. Outputs of calibrated hydrological/hydraulic models that are run for long simulation times can provide a vector layer of surface water that can be converted into a raster format at a resolution that is compatible with various remote sensing products. In particular, these outputs can be disaggregated to the spatial resolution that is suitable for compatibility with SAR data. Cian et al. (2018) [6] summarized 21 studies using SAR data for flood mapping and found only one study [20] that integrated SAR data with hydrological models. Using thresholding methods to extract water is ideal for rapid flood mapping [6,21]; however, this methodology is affected by sources of error that are typical in the use of SAR images, such as double returns from inundated vegetation and urban structures increasing backscatter in contrast with the water surface, dry and smooth bare soil exhibiting backscatter similar to that of water surfaces, and vegetation covering part of the flood area [6]. Image thresholding is simply defined as the masking of all flooded pixels with a radar backscatter lower than a certain threshold value [22,23]. In contrast, determining the optimal threshold value for extracting inundated areas remains a challenge. In this study, we developed a novel algorithm named Hammock Swing Thresholding (HST) to ascertain the optimal threshold required to extract robust water masks from SAR data. A calibrated 2D hydraulic model (HEC-RAS) provided accurate modeled inundation maps which were used to validate the thresholding adjustment to determine the SAR water areas using four SAR platforms, namely, Sentinel-1 (S1), ALOS-2 (AL2), TerraSAR-X (TSX), and RadarSAT-2 (RS2).

## 2. Materials and Methods

### 2.1. Study Site

Tam Nong district is located in Dong Thap province, Vietnam, in the lower Mekong River basin (Figure 1). The district is home to 220,000 people (2019 statistics), who live in an area of 459 km$^2$. The topography is relatively flat, with elevations ranging from 0 to 4 m above the mean sea level [24]. The climate of the region is influenced by the South Asian monsoon system, which is divided into two distinct seasons, i.e., the rainy season (May to November) and the dry season (December to April). Approximately 75% of the discharge (mean 14,500 m$^3$/s [25]) of the basin occurs in the rainy season [26]. The average annual humidity is 82.5%. The average rainfall range varies from 1170 to 1520 mm and is concentrated in the rainy season, which accounts for 90%–95% of the annual rainfall total.

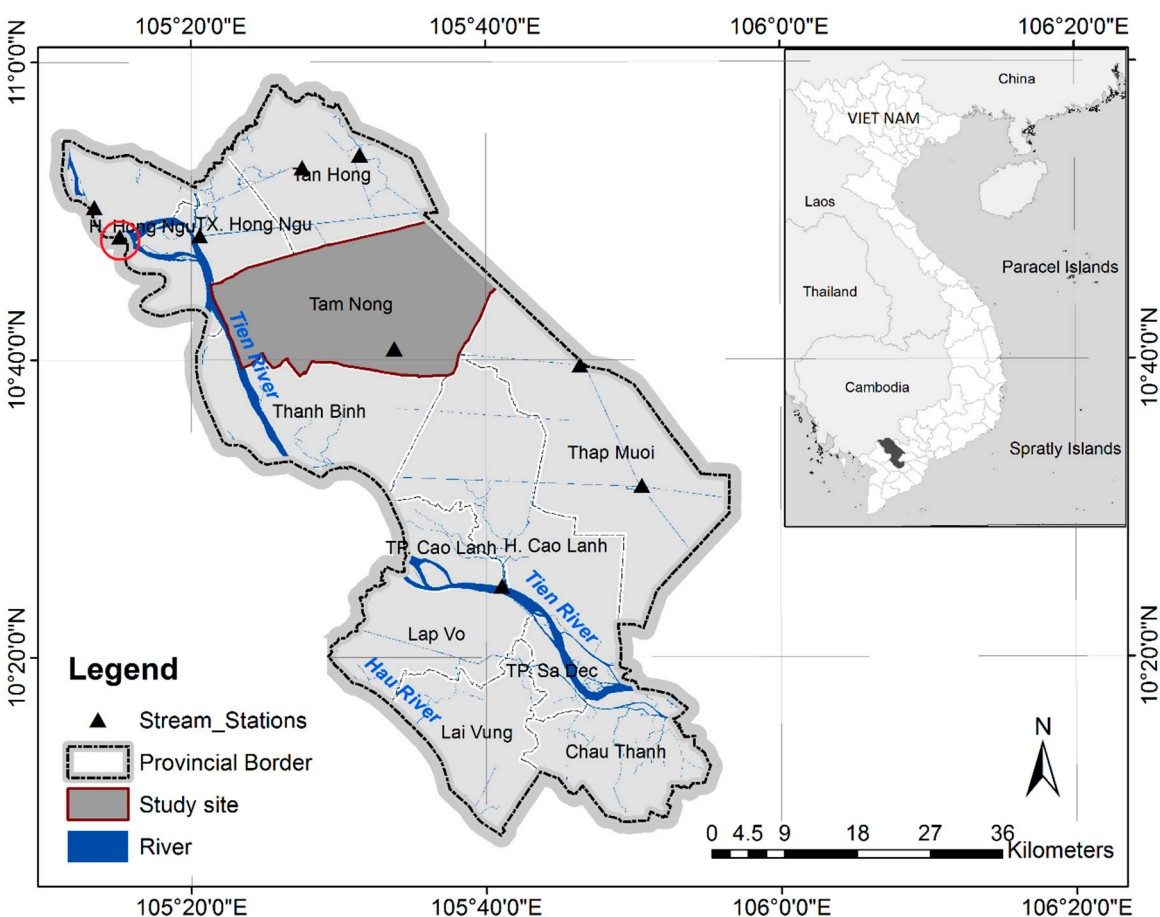

**Figure 1.** Site map of Tam Nong district in Dong Thap province (in the red polygon) located on the Vietnamese Mekong delta.

The flood season lasts from August to December and the highest peaks (4.58, 3.15, and 3.5 m at Tan Chau in 2000, 2010, and 2019, respectively) usually occur between the middle of September and the middle of October [27].

Vietnamese Mekong delta (VMD) flooding is one of the greatest influences on many aspects of human life, with both positive (providing fresh water and fluvial sediment) and negative aspects. Additionally, a rapid expansion of canal, dyke, and sluice infrastructure for different purposes has occurred, with intensive agricultural and aquacultural activities being promoted by the government's development strategy [28]. These factors could result in hydrological alterations in local flood behavior and affect the conventional agricultural culture.

Broadly, dams, sediment mining, and climate change effects led to significant changes in the hydrological regime, the environment, and various ecosystems [15,29], such as a decline in the flood season water discharge and annual sediment flux, water quality degradation, riverine aquatic biological communities, and fish assemblages [30]. The sediment discharge is −80 million ton/year and decreased over the last decades as a result of dam construction [31]. Consequently, concerns exist regarding dam development in the basin, which may adversely affect agriculture and fisheries [27].

*2.2. Methods*

2.2.1. Study Workflow

Our simplified study workflow is shown in Figure 2 and comprises two parts, i.e., SAR image processing and hydraulic inundation modeling. SAR image processing for both the pre- and during-flood images includes the fundamental steps of radiometric calibration, speckle filtering,

and terrain correction described in detail in other studies [5,6]; therefore, they are not discussed in this work. However, the thresholding approach based on matching the SAR-based water surfaces with hydraulic modeling water masks is the core process and is described further below. Although the thresholding approach for water extraction is commonly used for SAR images, we developed a new method named Hammock Swing Thresholding (HST) which fixed the threshold values to those that resulted in the best fit between the SAR and modeled water levels.

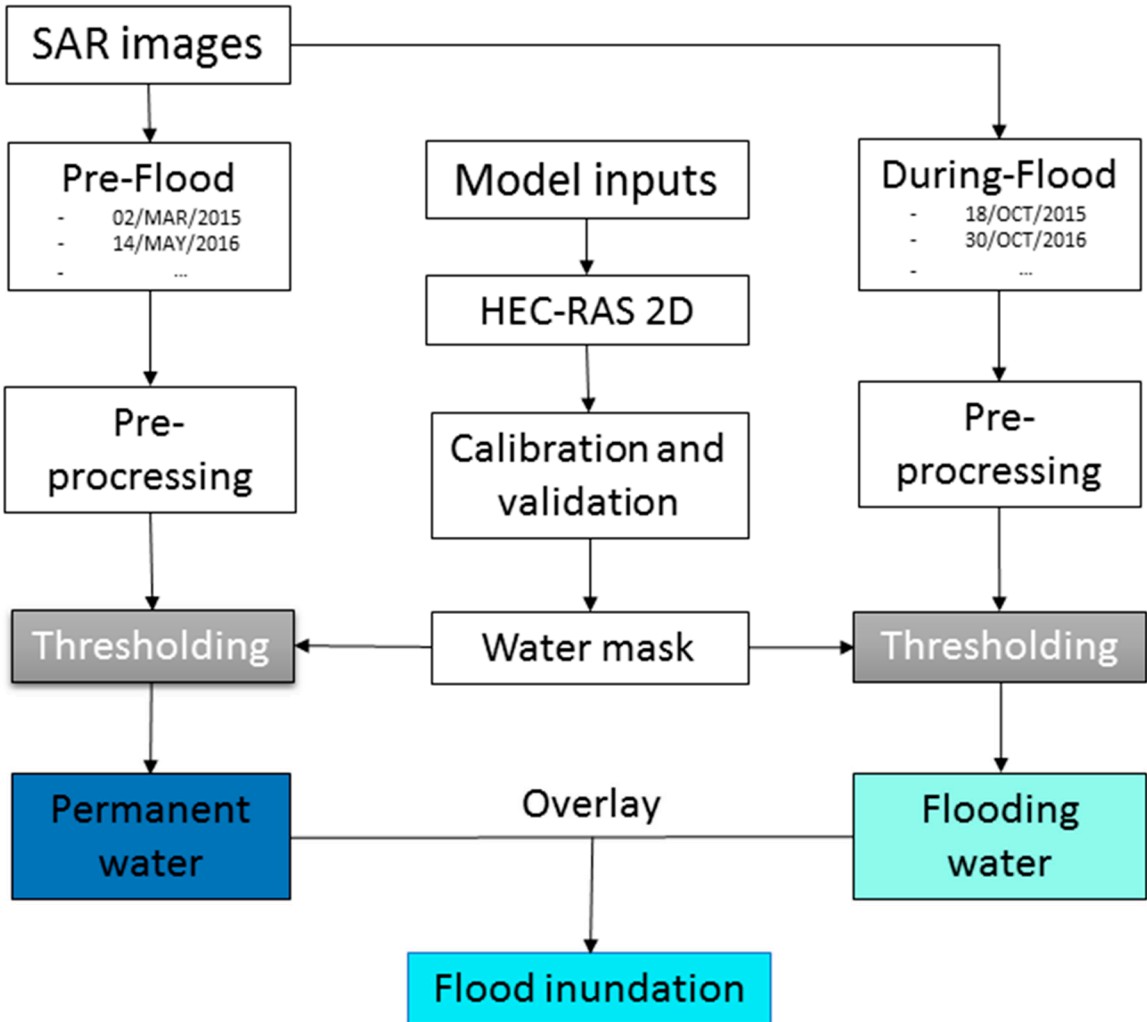

**Figure 2.** Flowchart of the methodology. SAR stands for Synthetic Aperture Radar and HEC-RAS 2D is two dimentional Hydrologic Engineering Centers River Analysis System model.

2.2.2. Inundation Hydraulic Modeling

- HEC-RAS 2D model setup

The HEC-RAS 2D model was set up for inputs of geometric data, flow area, mesh grid definition, breaklines, and boundary condition lines (Figure 3). Although the model geometry was reliant on the digital elevation model (DEM, the finer the resolution, the better), defining the mesh size and breaklines was crucial in terms of trade-off between the model details and the computational time and output storage. In this model, the terrain was flat and a very fine $5 \times 5$ m DEM was used; hence, we defined an irregular computational mesh grid (represented by back polygons) and defined breaklines for the main dykes. The flow area covered the entirety of the study site, thereby ensuring that the exported water surfaces mapped well with the SAR-extracted water masks. The unsteady flow data, including

the boundary conditions and initial conditions, were defined using the daily water level (WL) data presented in Section 2.3.1. It was noted that the times of the SAR acquisition sat within the HEC-RAS model simulation time.

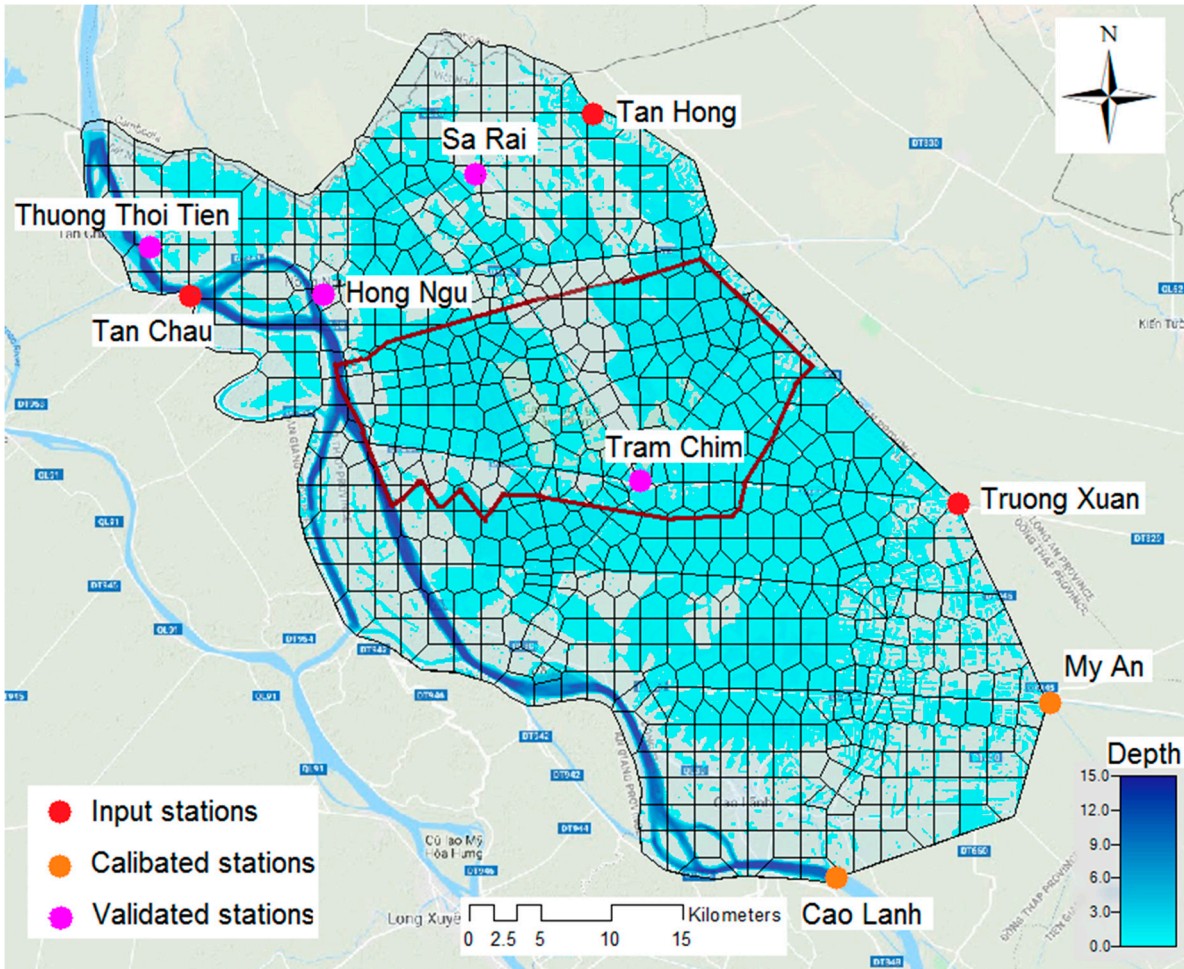

**Figure 3.** Dong Thap HEC-RAS 2D model setup with stream gauges marked.

Given the focus of this study was the presentation of the HST approach to calibrate SAR inundation extents rather than to describe the modeling procedures, brief summaries of the model setup and validation are provided below. References [12,32] provide more detailed descriptions of the model setup.

- Model parameterizations

The HEC-RAS model allowed users to generate geometric parameterizations with various options, including forcing mesh computation with user-defined point spaces, land cover to Manning's n coefficients adjustment (add constant values, multiply factors, set values, and replace values) [33] and enforce selected breaklines (enforcing mesh around them). The initial Manning's n roughness coefficient (N) was parameterized for nine land use/land cover (LULC) classes (Table 1). As the initial N values were taken from literature and calculated for other regions, adjustment was needed to account for the actual land use conditions of each class in the study site. Since the N values were highly dependent on a number of factors (such as surface roughness, vegetation, obstructions, etc.), estimating optimal initial values was based on ground truth information and the expertise of the modelers [34]. Optimal N values were considered to be crucial to shorten the computational time and support the calibration process within the HEC-RAS 2D modeling [35]. All of the layers (i.e., the terrain

and LULC maps) were projected in the Geographic Coordinate System_World Geodetic System_1984 (GCS_WGS_1984) projection.

**Table 1.** Manning's n roughness coefficients (N).

| Land use/land cover (LULC) | Default N | Calibrated N |
|:---:|:---:|:---:|
| 2-season rice | 0.060 | 0.015 |
| 3-season rice | 0.060 | 0.380 |
| Aquaculture ponds | 0.045 | 0.076 |
| Fruit trees | 0.400 | 0.750 |
| Mangroves | 0.035 | 0.037 |
| Residence | 0.068 | 0.760 |
| River and channel | 0.035 | 0.040 |
| Road | 0.032 | 0.780 |
| Wet grass | 0.035 | 0.045 |

- Model calibration and validation

Field water level measurements at Cao Lanh and My An (in orange; Figure 3) were used for calibration, whilst Hong Ngu, Sa Rai, Tram Chim, and Thuong Thoi Tien stream stations (Figure 3) were used for the model validation. The initial parameters were adjusted until the correlation between the observed and modeled water levels (based on coefficient of determination ($R^2$) and the Nash–Sutcliffe efficiency coefficient (NSE), where the ideal value was 1 and the accepted value was 0.65 [36], in the calibration and validation stages reached the accepted values. Visual checks were also performed to assess data bias.

### 2.2.3. Scattering, Polarization, and Local Incidence Angle Analyses

Three pre-thresholding processes were conducted to better understand the radar scattering mechanisms, the effects of different SAR polarizations, and the local incidence angles on each LULC class while considering open water (such as rivers, flooded rice paddies, and aquaculture ponds) and non-water (such as residences, 3-season rice, and roads) factors. First, backscattering in decibels (dB) of all of the SAR images was zonally, statistically analyzed to calculate the mean values of dB and the standard deviations for each LULC classes. As radar antenna are normally designed to receive different polarizations (where HH: horizontal transmit and horizontal receive; VV: vertical transmit and vertical receive; HV: horizontal transmit and vertical receive; VH: vertical transmit and horizontal receive), the added complexity of dual and four polarizations of the incoming electromagnetic waves necessitated tests to assess the effects of different polarizations on the radar scattering for the target geographical features. Finally, the local incidence angle (LIA), the angle between the normal vector of the backscattering element, and the incoming radiation vector formed by the satellite position and the backscattering element position [37] were varied from different SAR receivers. In various cases, LIA information was important for the distinct classification rules [38]; accordingly, we extracted the LIAs of Sentinel-1 (S1) and RadarSAT-2 (RS2) for comparison. All backscattering dB, polarization, and local incidence effects were performed using the zonal statistics of ArcGIS software version 10.2 of ESRI, USA by overlaying the SAR images (in decibels (dB)) or local incidence maps (in degrees) with the LULC map (see Sections 3.3 and 3.4 for results).

### 2.2.4. Thresholding

The simple concept underpinning the proposed hammock swing thresholding (HST) method was fixing the threshold values applied to the SAR images by minimizing the total absolute values of the difference layer between the SAR and HEC-RAS water extent output rasters (Figure 4). The HST name was given because the shapes of the HST graphical representations were similar to the shapes of a hammock's swinging actions, which are typical in some Asian cultures. In the flood-extent maps

extracted from both SAR and HEC-RAS, we defined water (pixel value of 1) and no-water (pixel value of 0) pixels. Pixel values of the difference layer were 1, 0, and −1, with a perfect match being depicted by a layer where all pixels had a value of zero.

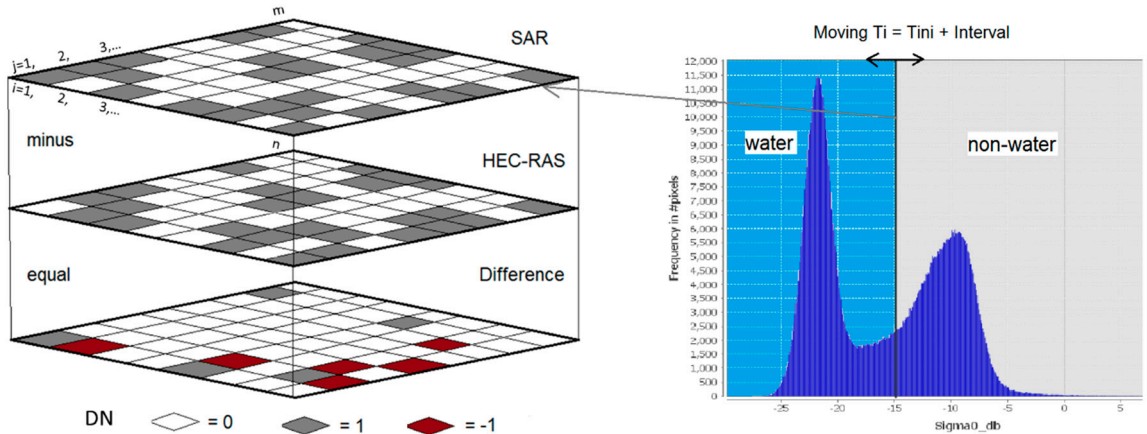

**Figure 4.** Graph of the Hammock Swing Thresholding (HST) concept determining an optimal SAR threshold of water masking. HEC-RAS stands for the Hydrologic Engineering Centers River Analysis System model. DN is a digital number, where 0 represents a perfect match between SAR and the model outputs, 1 represents underestimation of the water extent by SAR, and −1 represents overestimation of the water extent by SAR. Ti is the threshold dB value i, Tini is an initial threshold, and m and n are numbers of pixel columns and rows, respectively, that vary in different types of SAR images.

Mathematically, the minimum or optimal absolute value of difference ($VD_{opt}$) was defined in Equation (1), such that

$$VD_{opt} = \min\left(\sum_{Ti=1}^{n} \left|DN_{SARij} - DN_{HECij}\right|\right), \tag{1}$$

where $VD_{opt}$ is the optimal absolute value of difference between two compared raster layers, $DN_{SARij}$ and $DN_{HECij}$ are the pixel values of SAR and HEC-RAS water masks, respectively, and $n$ is the number of threshold adjustments ($Ti$) for the SAR images.

The $DN_{SARij}$ at threshold $i$ was calculated from the sigma nought decibel band ($\sigma^0_{dB}$) using the band math tool with various expressions from the Sentinel Application Platform (SNAP) processing software. The $\sigma^0_{dB}$ values were solved using Equation (2).

$$\sigma^0_{dB} = 10 \times \log_{10} \sigma^0, \tag{2}$$

$$\sigma^0 = \frac{DN^2}{A^2_{dn}} \times \frac{1}{G^2_{eap}} \times \left(\frac{R}{R_{ref}}\right)^3 \times \sin(\propto), \tag{3}$$

where $\frac{1}{G^2_{eap}}$ is the elevation antenna pattern (EAP) correction (2-way), $\left(\frac{R}{R_{ref}}\right)^3$ indicates the range-spreading loss (RSL) correction, $A^2_{dn}$ is the product final scaling from an internal single look complex (SLC) to a final SLC or ground-range detected (GRD), $\propto$ is the local incidence angle, and $DN^2$ is the average product intensity with a value of 22,142.71.

In the graphical evaluation phase, we presented the relationship between the different residual $RE_{SAR-HEC}$ on the Y axis of a graph, where

$$RE_{SAR-HEC} = \sum_{i=1}^{n} \left|DN_{SARij} - DN_{HECij}\right|, \tag{4}$$

and percentages of the mutual flooded pixel values $P_{SAR\&HEC}$ were solved using Equation (5) in both the SAR and HEC-RAS flood images (on the y axis), with the SAR decibel value $\sigma_{dB}^0$ being solved by Equation (2) (on the x axis).

$$P_{SAR\&HEC} = \frac{\sum_i^j DN_{SARij} - RE_{SAR-HEC}}{\sum_i^j DN_{SARij}} \times 100 \tag{5}$$

The HST algorithm procedures are presented in Figure 5, where the SAR images processes were performed semi-automatically using the batch-processing tool in the SNAP software environment. The HEC-RAS software provided options to export outputs both in vector and raster formats. It was recommended to output vector files for the HEC-RAS water masks, as this gave users the capacity to convert them to the raster format, which theoretically had identical pixel sizes to the corresponding SAR images. Spatial statistics were applied using the raster calculator and summary statistics tools using the ArcGIS 10.2 software.

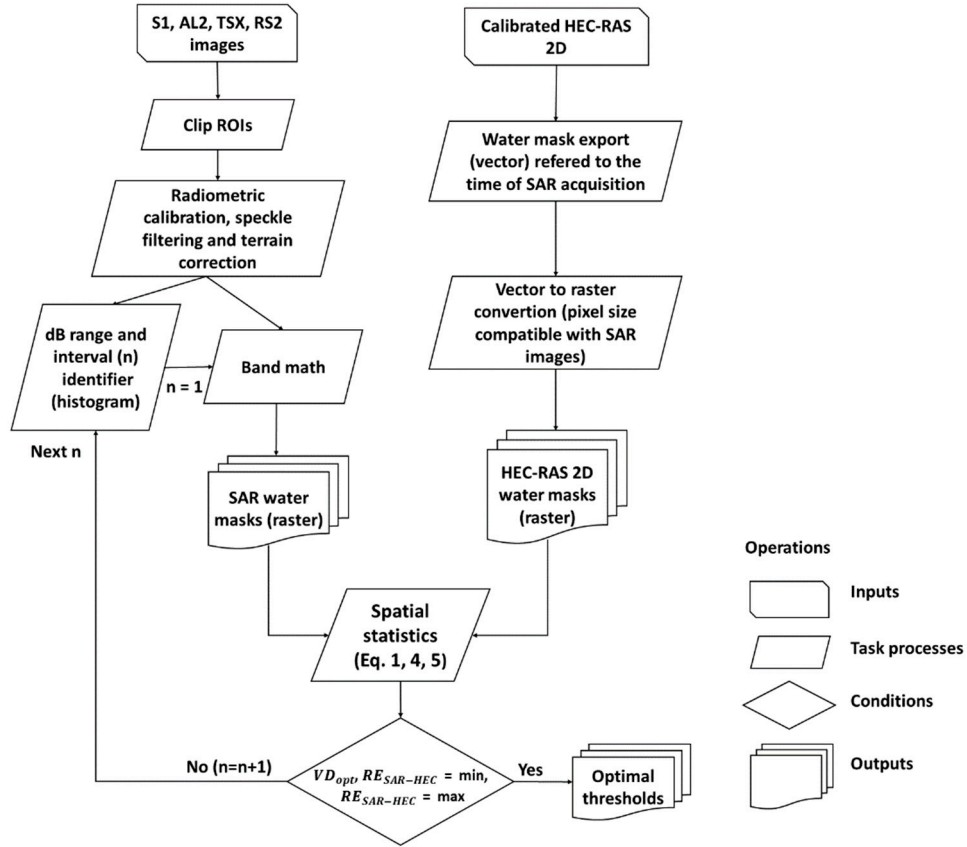

**Figure 5.** Schematic of the HST algorithm to obtain optimal thresholds. S1, AL2, TSX and RS2 stand for Sentinel-1, ALOS2, TerraSAR-X and RadarSAT-2, respectively. ROIs are regions of interest. Equations (1), (4), (5) are the equations defined in the above. n is the number of threshold adjustments defined in the Equation (1). The $VD_{opt}$, $RE_{SAR-HEC}$ and $P_{SAR\&HEC}$ are defined in the Equations (1), (4) and (5).

2.2.5. Generating Flood Maps and Evaluation

Inundation maps were generated by "overlaying" the pre- and during- flood water masks using the raster calculator tool in the ArcGIS software. The pre-flood images were assigned a value of nine for dry pixels and 10 for wet pixels, while the during-flood images were assigned values of zero for dry pixels and two for wet pixels. By subtracting the pre-flood images from the during-flood image, a flood map was generated which identified areas of flood inundation (pixel value of 7), permanent

water (pixel value of 8), dry areas (pixel value of 9), and the transition from wet to dry (pixel value of 10). This task was repeated for all SAR images. The resulting maps are presented in Section 3.6.

Uncertainty surrounding the extracted water surface using thresholding approaches resulted from various sources, such as the input data (image spatial resolutions, geometric distortions, radiometric noise (speckles)) and the algorithm structure (intervals, time-matching between the HEC-RAS 2D model, and imaging times of the SAR data). Here, we report only the best matches between the SAR-based water surfaces with the water surface from the calibrated HEC-RAS model, and provide some comparisons to additional estimated optimal thresholds from previously published results.

### 2.3. Data

#### 2.3.1. Data for the HEC-RAS 2D Model

The HEC-RAS 2D model required topography data, land use/land cover (LULC) information, water level (WL) measurements, and Manning's n values for the different LULC classes. A high-resolution digital elevation model (DEM) of 5 m spatial resolution produced by Vietnam Natural Resources and Environment Corporation (VINANREN) under the Ministry of Natural Resources and Environment (MONRE) in 2014 was provided for the model geometry setup (Figure 6). The DEM, which had vertical errors of less than 0.17 m, was generated using photogrammetric techniques and airborne images following the guidelines to produce DEMs and topographic maps from MONRE. At VINANREN, the airborne images were processed using the digital photogrammetric workstation of Intergraph Corporation (Huntsville, AL, USA), which was called ImageStation Z1.

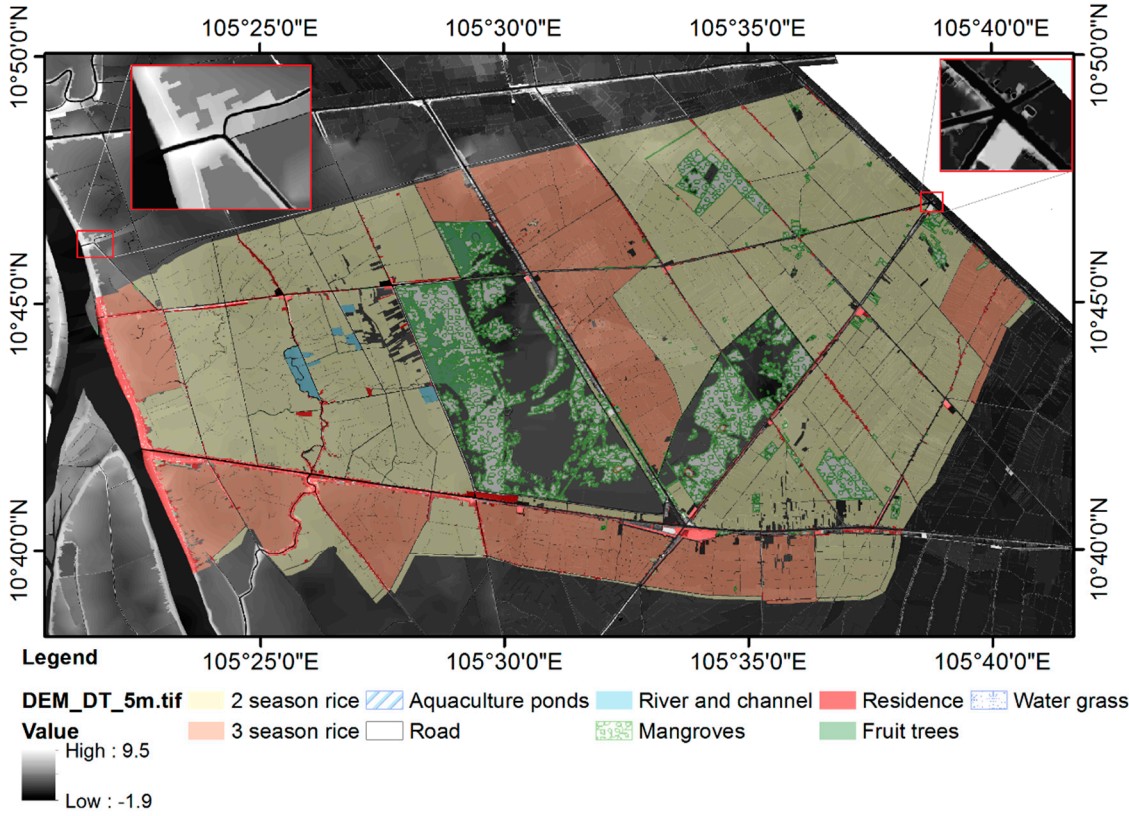

**Figure 6.** Land use/land cover (LULC) map of Tam Nong district, which was updated in October 2018 and overlaid with the 5 m digital elevation model (DEM).

A LULC map with 9 classes consisting of 2- and 3-season rice crops, aquaculture ponds, fruit trees, mangroves, residences, rivers and channels, roads, and wet grass provided by MONRE was

employed to analyze the LULC information of the model. The LULC map was validated and updated in October 2016, mainly in regard to the rice-farming classifications and inundated vegetation (Figure 6). The default Manning's n coefficients from Engman (1986) and Bunya et al. (2010) [39,40] for overland and Barnes et al. (1967) [41] for water bodies were initially assigned to the relevant LULC layers and adjusted during the calibration process.

Finally, hydrological data for the boundary conditions of the model were collected from Tan Chau and Cao Lanh stream-gauges (Figure 7). Specifically, a 21-year, (1996–2016) daily WL dataset was included for the model setup. Other in situ daily water level recordings from 2000 to 2015 were derived from four stations, namely, Hong Ngu, My An, Truong Xuan, and Tram Chim, for model calibration and validation. The new stations, Sa Rai and Thuong Thoi Tien, were installed in 2015 and measured in-field water levels which were employed for validation of the local flood stage.

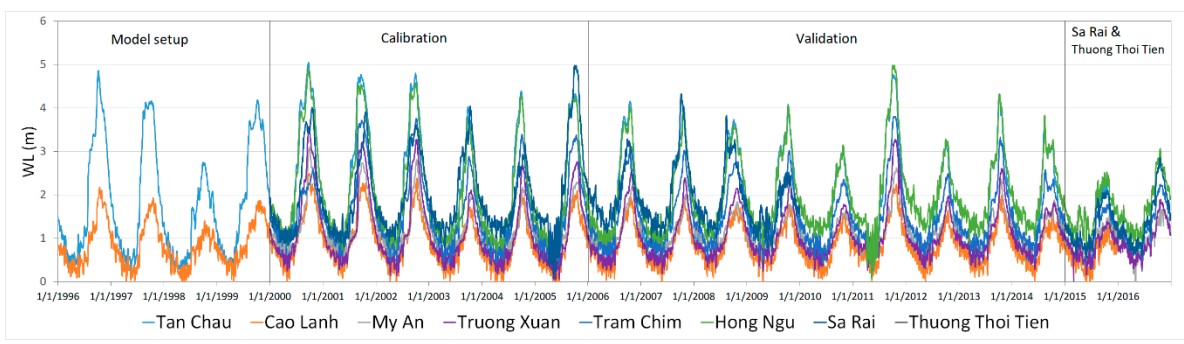

**Figure 7.** Hydrological data for the Dong Thap HEC-RAS model. WL: water level.

### 2.3.2. Remote Sensing Data

We used four different SAR datasets from four space agencies, namely, the European Space Agency (S1), Japan Aerospace Exploration Agency (AL2), German Aerospace Center (TSX), and Canadian Space Agency (RS2). The metadata for the images used are summarized in Table 2. It was noted that the acquisition time was chosen based on the flood (September–November) and dry (February–April) seasons in the Vietnamese Mekong delta, as based off of water levels at Tan Chau (marked by the red circle in Figure 1) [38]. The satellite images acquired in the dry season were used for reference.

**Table 2.** Fundamental information of the used SAR images.

| Sensor Band Imaging Mode | Pass Polarization | Date of Acquisition | Incidence Angle | Status | Ground Resolution ¥ |
|---|---|---|---|---|---|
| Setinel-1 C IW | Descending VV VH | 2014/10/18 2015/02/03 | 30.86–45.99 30.86–45.99 | Flood Ref. Ͳ | 10 m |
| ALOS2 L SGI | Ascending HH HV | 2015/10/23 | 40.29 * | Flood Ref. (Ͳ) | 25 m |
| TerraSAR-X X ScanSAR | Descending HH | 2010/11/04 | 31.8–40.5 | Flood Ref. (HEC) | 8.25 m |
| RardarSAT-2 C Wide Fine | Ascending VV VH | 2016/10/30 2016/05/14 | 30.6–39.5 30.6–39.5 | Flood Ref. | 5.2 m |

* at the scene center; ¥ pixel space after geometric terrain correction and re-projection; Ref.: reference; Ͳ Sentinel-1: a reference for both Sentinel-1 and ALOS-2 flood extraction; Ref. (HEC): reference water mask extract from the HEC-RAS 2D model. Letters C, L, and X were used to name the SAR bands; IW: Interferometric Wide imaging mode; SGI: standard geocoded image. Letters V, H are vertical and horizontal, respectively and. Coupled letters of VV, VH, HV and HH indicate SAR cross-polarized polarizations.

## 3. Results and Discussions

### 3.1. HEC-RAS 2D Calibration and Validation

HEC-RAS 2D was set up for approximately 90% of the area of Dong Thap province and was calibrated for water levels at Cao Lanh and My An stations for the ten year period of 1996 to 2005.

The level of agreement between the simulated and the observed daily WLs was evaluated using the coefficient of determination ($R^2$) and the Nash–Sutcliffe simulation efficiency coefficient (NSE), both of which were 0.86 for Cao Lanh and 0.78 for My An, compared to the accepted value of 0.65 [42,43] (Figure 8).

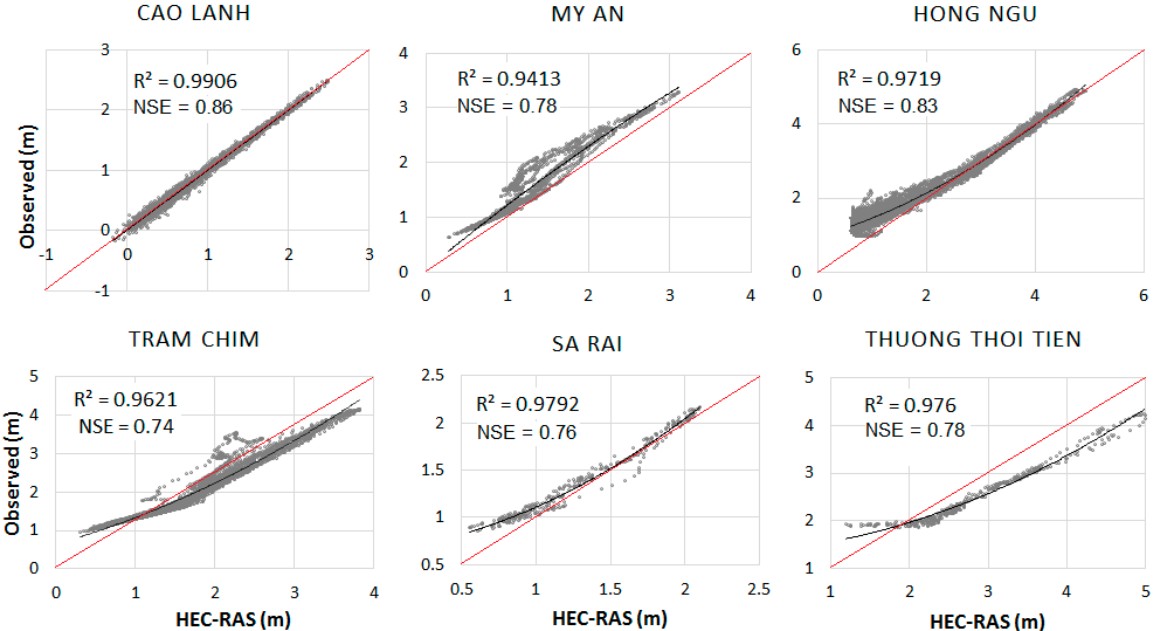

**Figure 8.** Results of calibration and validation for the HEC-RAS 2D Dong Thap model.

Validation was performed using in situ measured daily WLs for the period 2006–2016 for Hong Ngu and Tram Chim, and the 2015–2016 and 2016 periods for Sa Rai and Thuong Thoi Tien stations, respectively.

As the Sa Rai and Thuong Thoi Tien were newly installed stations, the observed data were only available for these short periods of time and were compared with the model simulated data independently. Figure 8 shows the modeled HEC-RAS water levels versus the observed water level data. As can be seen, the data were plotted close to the 1:1 line in all cases and displayed acceptable values for $R^2$ and NSE, however the lowest NSE value (0.74) at Tram Chim station was lower than the acceptable value of 0.78. Therefore, the model was stable not only in the calibration time period, but also in validation periods even at the weakest point (Tram Chim, which was far from the calibrated stations) of the model. Since the model was deemed accurate, the extracted water surfaces were used for SAR image (Table 2) thresholding.

### 3.2. Scattering Analysis for Pre-HST Thresholding

Our mean dB value results (ordered from low values on the left to high values on the right) and standard deviations (STD) of the nine major LULC classes and predictive water thresholds (the blue line) for the flood S1, AL2, TSX, and RS2 images (listed in Table 2) are shown in Figure 9. The 2-season rice, rivers and channels, and pond classes received low dB values from the SAR images and were considered to be water surfaces (blue boxes) under the uneven predictive thresholds. Conversely, the residence, road, and fruit tree categories were depicted as dry (grey boxes), with dB values well above the thresholds. Although the highest dB values were found for mangrove (RS2) and grass (S1 and TSX), we assigned these cells as wet or dry areas, since the mangroves and grass could cover water surfaces; the mangroves depicted in the AL2 data (with the highest pixel values) were assured to be inundated with vegetation when the SAR data was collected. The 3-season rice in the S1 and TSX images was erroneously classified as water or dry layers (boxes S), as some plots' values were low

(maybe irrigated) while others were high (dry). The standard deviations (the vertical bars) showed the ranges of the dB values which were different not only between classes, but also between the SAR images. The differences in mean dB values for S1, AL2, and TSX were all around 10 dB, which possibly revealed the consistency of the decibel (dB) values recorded by the different SAR sensors and therefore was applicable for image thresholding. As seen in Figure 9, the mangroves in the AL2 image were predicted to be inundated where the radar pulse was double-bounced; hence, we received a mean value of 75 dB, which was even higher than the dB value of the residence class. However, the mangrove areas captured in S1, TSX, and RS2 were not identified as flooded areas (orange boxes) due to the dB value being very close to the dB values of the known dry areas.

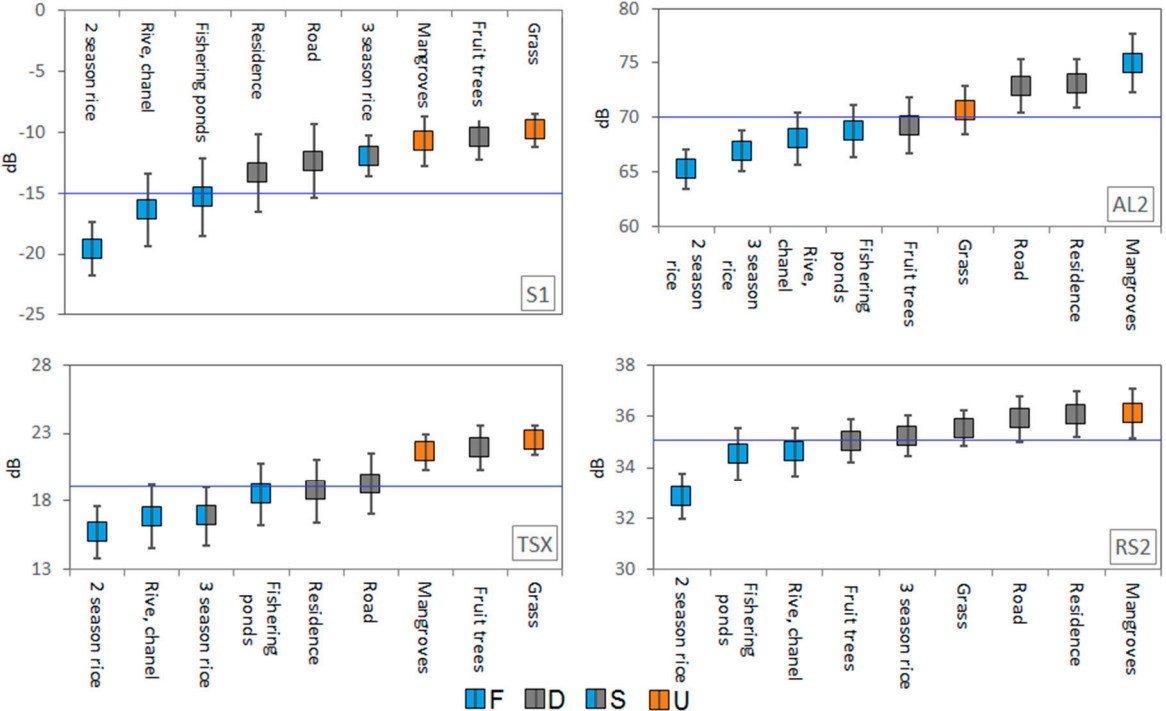

**Figure 9.** Mean sigma0 dB values and standard deviation (STD) of the major LULC classes in Tam Nong. F: flooded; D: dry; S: semi-flooded (some plots were underwater whereas other plots were dry); U: remains unidentified.

*3.3. Polarization Comparison*

Comparisons of the received scattering of the dB values of different polarizations (HH, VV, HV, and VH) designed for S1, AL2, and RS2 in the context of different LULC classes, which were divided into dry and flooded areas, are shown in Figure 10. The TerraSAR-X image was acquired for the HH polarization only, therefore, a comparison of its polarizations was not possible. Wide gaps in the dB values between the VH and VV bands of S1 and the HH and VV bands of RS2 (both recorded at the C band) were observed. However, the standard deviations of the RS2 VV mean values (about 2 dB) were much lower than those of the RS2 HH band (approximately 8 dB). Although the wide gap in the dB values showed the difference in the range of values between the two polarizations, the mean value lines of S1 and RS2 showed a general rise in mean values from wet areas (rivers, 2-season rice, and 3-season rice) to dry areas (fruit trees, grass, roads, and residences). In contrast, there was a minimal difference between AL2 HH and HV polarization (recorded at the L band) with the exception of the river, channel, and mangrove classes. These exceptions could be explained by the effects of vegetation cover on the water bodies (both river and mangrove) where the double-bounce effect that took place with different polarizations was not equal. There is still a need for further research into why the double-bounce of the L band observed for VV was lower than that observed for HH.

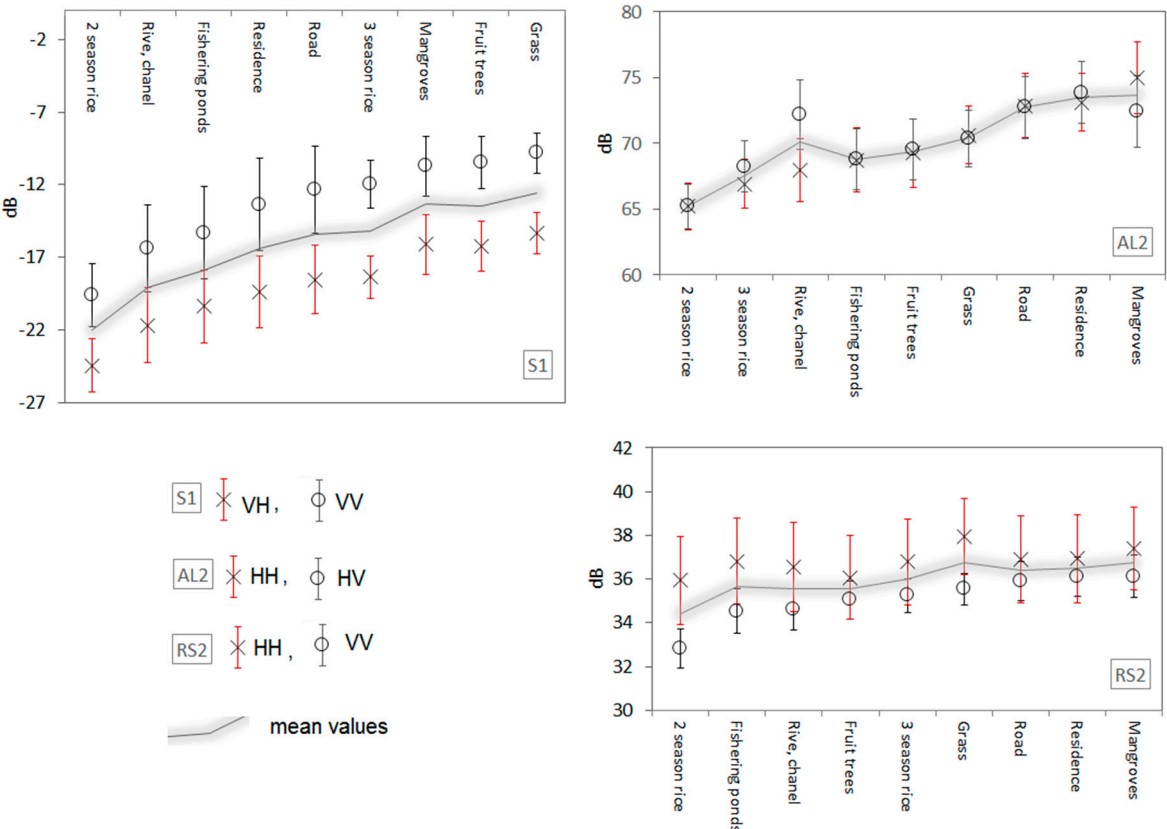

**Figure 10.** Comparison effects of polarizations on mean dB values of each LULC class (the glowing lines represent the mean values of corresponding pair polarizations).

### 3.4. Local Incidence Angle Effects

The mean local incidence angles (LIA) of the nine LULC classes exported from the four SAR datasets were summarized and ordered in increasing size from left to right for S1. The LULC order was kept unchanged for the rest of the images, as shown in Figure 11. The colored boxes indicated rough predictions of wet (blue), dry (grey), and unidentified areas (dark orange). The statistics showed that while the open water bodies were calculated to have smaller mean LIAs compared to dry areas (roads, fruit trees, and residences) in the descending results of S1 and TSX, the reverse trend depicted a decrease in mean LIAs for the AL2 and RS2 data (ascending). However, a stable trend was observed when a higher resolution DEM was used (1 s degree resolution) from the Shuttle Radar Topography Mission (SRTM1; white boxes), thereby generating a slightly larger (0.1 degree) LIA than that obtained when using the 3 s SRTM DEM (SRTM3) in other LIA extractions (colored boxes). In addition, the half STDs from the SRTM1 LIAs (~1.5) were almost double those from the SRTM3 (~0.75) in all of the images. Since the STDs were much larger (15 times larger for SRTM3 and 30 times larger for SRTM1) than the differences between the mean LIAs in each LULC type, there was a wide range in the LIA mixtures for all of the LULC classes; hence the extracted LIA information was deemed inapplicable for water extraction.

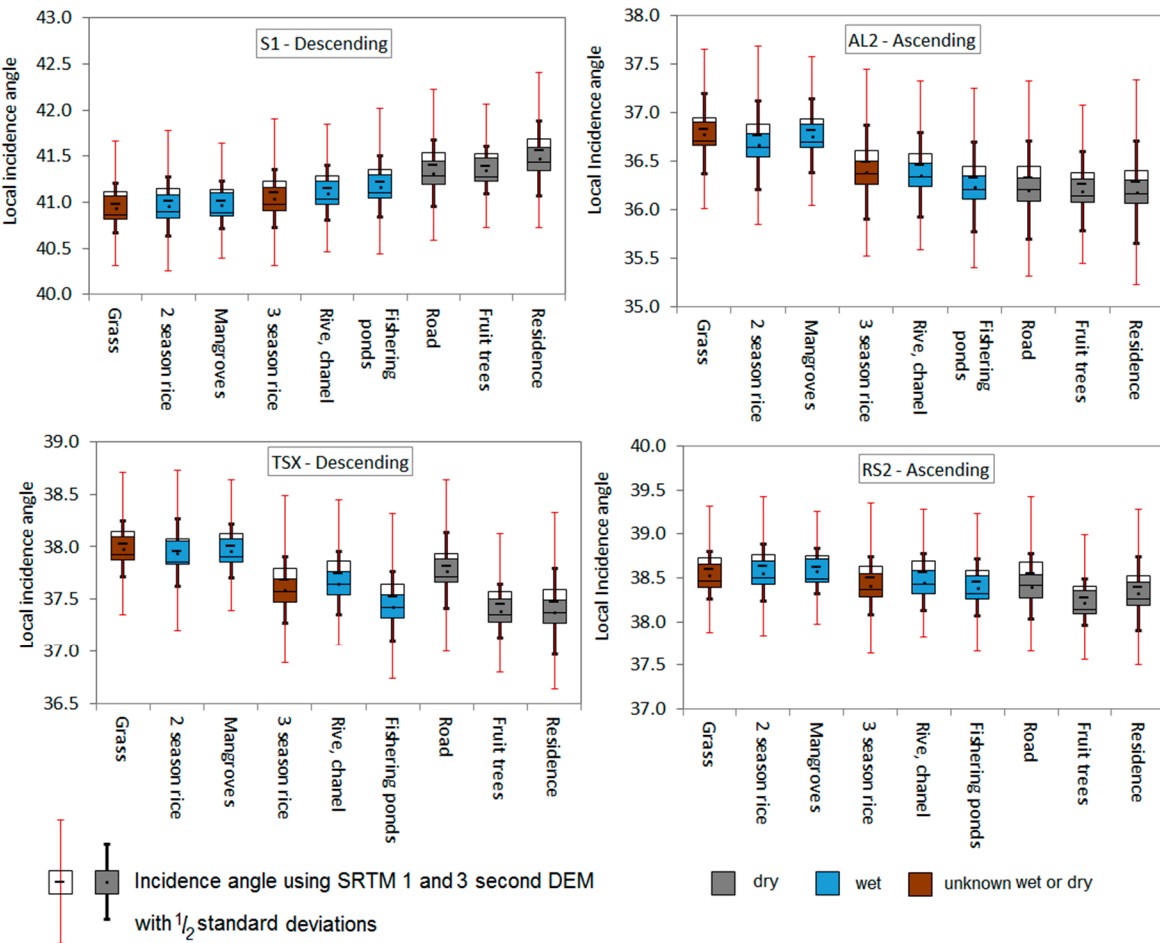

**Figure 11.** Relationship between local incidence angles and land use types with effects of DEM resolution (1 and 3 s Shuttle Rada Topography Mission (SRTM)).

*3.5. Graphical HST Optimization*

Figure 12 indicates the residual (RE) (calculated in Equation (4)), shown by black dotted lines, the percentages (P) of mutual flooded pixel values in the SAR and HEC-RAS modeled water masks (solved by Equation (5)), shown in black continuous lines, and the optimal threshold values where the REs were at their lowest point and the Ps were at their peak, as marked by the blue rectangles (maximum agreement) and circles (minimum difference). The differences and the agreement between the Sentinel-1 and calibrated HEC-RAS water surfaces are indicated in Figure 12A, with wide ranges of RE (120–220 thousand pixels) and P (60%–88%). However, the slope of the curve approaching the optimal value of 88.26% or dB threshold of minus 15 was flat, which meant that the Hammock needed to swing further and longer to approach the optimal spots. This case was similar to the AL2 image thresholding (Figure 12B), however the P was lower than S1's P by about 2.3%.

In contrast, the RE and P values of the TerraSAR-X and RadarSAT-2 HST variations covered a narrower range (Figure 12C,D). Furthermore, the slopes of the curves of the TerraSAR-X and RadarSAT-2 HST remained steep and the peaks of the curves were easier to determine than by using S1 and AL2 images. Figure 12 also shows the number of RE pixels (in the thousands) increasing sharply when the finer spatial resolutions of the SAR images of 25 m (AL2), 10 m (S1), 8.25 m (TSX), and 5.2 m (RS2) were used. Water extraction thresholds were different from region to region, however we found correlations with outcomes of previous studies, such as Cohen et al. (2016) [8] and Uddin et al. (2019) [44], who estimated S1 water thresholds of −15.6 dB (Finland), and −22.4 to −12.9 dB

(Bangladesh), respectively, compared to an HST of −15 dB. Additionally, Martinis et al. (2009) [3] simulated a TSX water threshold of 22.6–29 dB (UK), while the HST provided 21 dB.

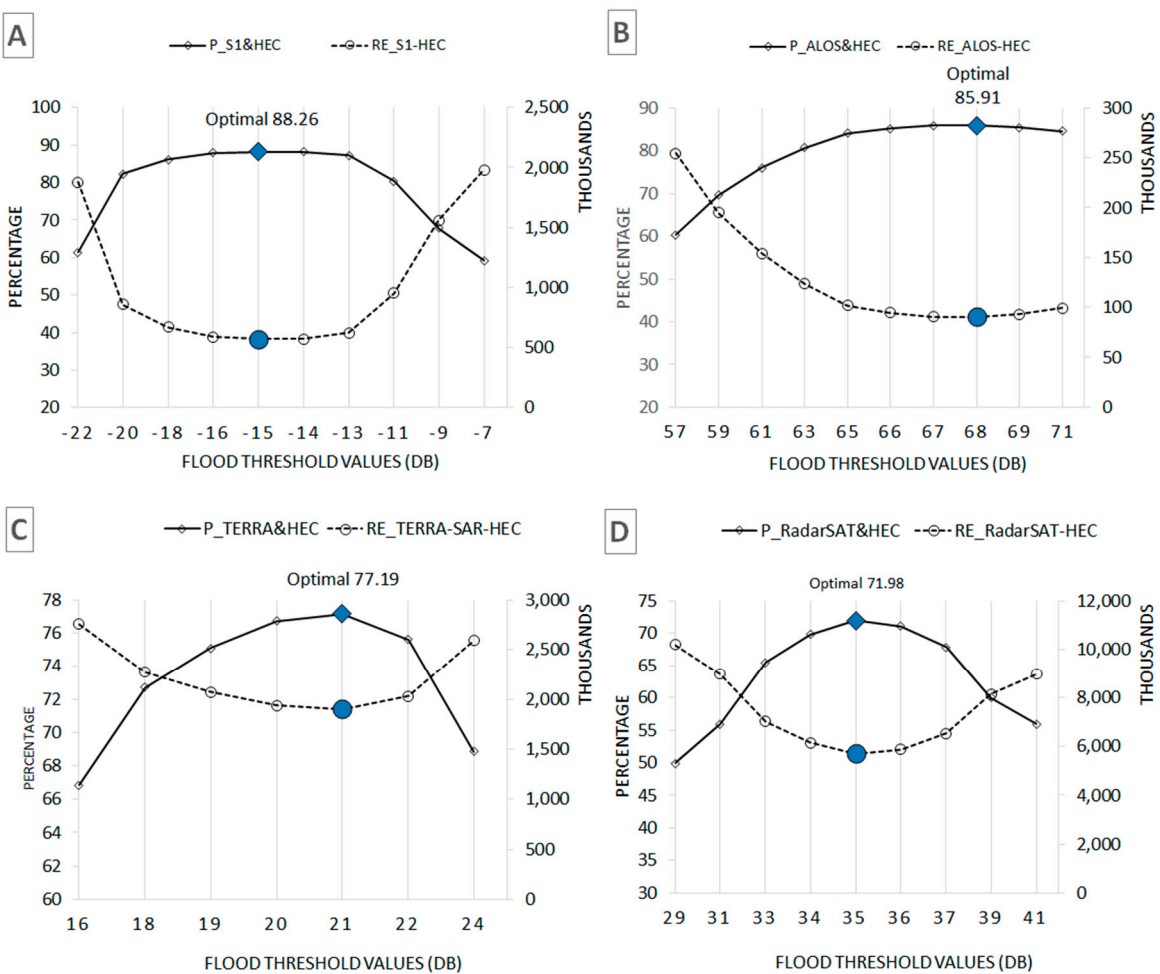

**Figure 12.** Graphical presentation of HST for (**A**) Sentinel-1, (**B**) ALOS PALSAR, (**C**) TerraSAR-X, and (**D**) RadarSAT-2 thresholding images of water extractions.

The agreement between the SAR and the hydraulic model flood extraction was also based on the model computational interval (a day for the Dong Thap model). In other words, determination of the time of the model flood extraction that approached the time of the SAR image sensing caused better agreement to be achieved. This was important for areas where flood duration and recession were rapid. There were differences between the optimal thresholds of the four SAR images, which may be linked to the noise (speckles) and resolution of the images. Despite all of the images being filtered to reduce the noise, finer resolution images contained more noise within the pixels than coarse images (a reduction in resolution declined speckles by multilook (3 × 3)) [45]. There was no noise observed when converting the HEC-RAS vector water to the raster format. In general, this may explain why the number of different pixels in RS2 was greater than S1 and the RS2 optimum was lower than S1.

*3.6. SAR Flood Maps*

Figure 13 shows the resulting inundation maps of the Tam Nong district extracted (with optimal thresholds) from four different types of SAR images all acquired in the flooding seasons (Q4), with permanent water defined as those pixels inundated in the dry seasons (Q2) of the years 2010 (TSX), 2014 (S1), 2015 (AL2), and 2016 (RS2). There were some sunken areas which displayed flooded water in all cases, however, the most mutual permanent water surface appeared between the S1 and AL2

maps compared to the TSX and RS2 imagemaps. It made less sense to compare flooded, permanent water from SAR images acquired at different times with different flood stages (marked by the red crosses on the graphs).

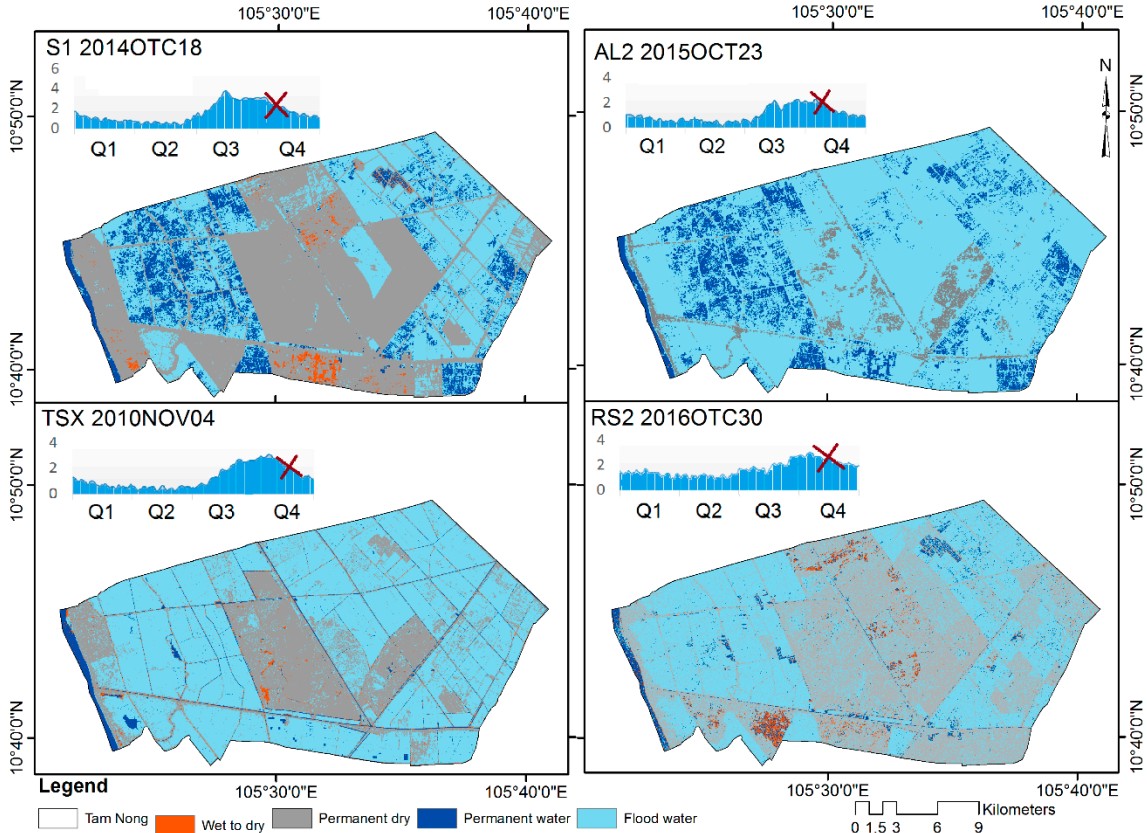

**Figure 13.** SAR remote sensing-based inundation maps. Flooded areas are shown in light blue, permanent water (i.e., appearing in both flooded and dry season) is shown in dark blue, and the orange pixels represent areas which turned from wet in the dry season to dry in the flooding season. Q stands for the year quarter.

Despite this, the finer spatial resolutions of 8.25 m and 5.2 m from the TSX and RS2 flood maps were much more detailed, revealing even the presence of small channels on the TSX and RS2 maps. These channels were mostly classified as dry pixels on the S1 and AL2 extractions. We also found some unexpected areas (in orange) which were covered by water in the dry season but appeared dry in the flood season. These observations may have been due to the SAR image acquisition times when some irrigation activities were active during the 3-season rice crops. The orange pixels located in Tram Chim national park (with no agriculture) were a result of background noise.

Given the resulting flood maps relied on the outputs of the well-calibrated HEC-RAS 2D model, the error bounds from the model were considered to be lower than satellite-born uncertainty. However, the uncertainty levels of the hydrological/hydraulic modeling flooded areas were up to ±30%. Hence, in this study we attempted to calibrate the HEC-RAS model elaborately, even though the main study focus was the optimization of the SAR-based water extraction thresholding. In addition, as there are many hydrological/hydraulic models available, choosing appropriate model types and scales to generate reliable results of final satellite-based flood inundation maps was imperative, as supported by Prestininzi et al. (2011) [46].

## 4. Conclusions

Calibrated hydraulic modeling outcomes with very fine spatial representations (in vector format) that are compatible to a wide range of spatial resolutions (down- and upscaling) are useful to determine optimal SAR image thresholds. As remote sensing (RS) data are acquired at certain times of the day with different data collection intervals, the collection of suitable reference data for calibration and validation of RS-based products is challenging. Hydrological/hydraulic modeling outputs are considered to mitigate this data gap.

In SAR image processing, the pre-process is considered crucial not only for further mandatory tasks, but also to extract useful information on radar scattering distribution over different types of geographic features. SAR sensors provide users with alternative polarizations and received scattering sensitivities to water surfaces and dry areas that may be dissimilar between polarizations and SAR sensors. Hence, choosing polarizations or ratios of them (e.g., HH/VV) for particular applications requires great care.

There are a wide range of applications using the thresholding approach to exploit SAR data for water extraction and flood mapping, as described by [3,45,47], as it is simple, yet robust, and particularly applicable for rapid and automatic flood mapping. Nevertheless, determining optimal thresholds is challenging [21]; hence, we proposed an "asymptotic approach" called Hammock Swing Thresholding (HST) that iteratively estimated SAR scattering dB thresholds as they approached the optimal values for four different SAR datasets. We asserted that the thresholds estimated were useful for rapid flood mapping applications, even for data-heavy applications and systems, thereby proving that these time-series flood maps can support flood monitoring and management. The optimal thresholds defined in this work were often not constant for every region. Thus, site-specific optimization is required, and uncertainties linked to SAR observations (e.g., speckles, low resolution) and ground perturbations, such as vegetation and terrain, must be quantified and accounted for [48].

**Author Contributions:** Conceptualization, C.R.H., N.H.Q., N.M.H., and V.A.T.; data collection and analysis, L.T.T.H., N.D.A., N.H.Q., and N.M.H.; writing—original draft preparation, D.T.D., D.T.T., L.T.T.H., N.H.Q., and V.A.T.; review and editing, C.R.H., N.H.Q., D.T.T., N.D.A., and V.A.T.; supervision, C.R.H. and V.A.T.

**Funding:** This research was funded by the Space Science and Technology Program, supported by Vietnam Academy of Science and Technology (VAST), project number VTUD.12/17-20.

**Acknowledgments:** Pham Viet Hoa at Ho Chi Minh City Institute of Resources Geography, VAST and Nguyen Vu Giang at Space Technology Institute, VAST for providing the DEM and the in situ water levels for the Dong Thap HEC-RAS model. The authors thank two anonymous reviewers whose comments substantially improved the paper.

**Conflicts of Interest:** The authors declare no conflict of interest.

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
