# Peer review of "Hydrological/Hydraulic Modeling-Based Thresholding of Multi SAR Remote Sensing Data for Flood Monitoring in Regions of the Vietnamese Lower Mekong River Basin"

_water, doi:10.3390/w12010071_

Round 1

Reviewer 1 Report

This work presents a case study in Vietnam Lower Mekong River Basin with the aim to generate flood inundation maps using the proposed Hammock Swing Thresholding method.

It uses a calibrated hydrological model (HEC-RAS) coupled with the outputs of the extracted water masks that are iteratively calculated using different db thresholds. Four different SAR platforms were tested. Each platform is of a different band with various polarizations. For the setup of the hydrological model a good quality topography data file (DEM) is required, as well as land cover (LCLU) information, water level measurements and the friction coefficients (Manning n) of the land cover classes. For the first step basic pre-processing performed on both pre and during flood SAR images. Then SAR-based water surfaces, both the pre-flood and the during-flood are matched with the hydrological modeling water masks. Modifying the threshold, which separates water from non-water pixels, allows the estimation of the optimum threshold that gives the minimum absolute difference between the calculated SAR water masks and the HEC-RAS water masks.  Eventually, the tuned masks are overlaid to form the flood inundation mask.

The idea that for the calculation of the water masks, many aspects of the morphology of the areas are taken into consideration, is credible.

When comparing the calibrated value to the default ones, there are times that the variation is big. The “Road” coefficient is changed from 0.032 to 0.78, the “Residence” from 0.068 to 0.760. Are such deviations common?

The graphical evaluation with the percentage of the mutual flooded areas and the different residual (between SAR and HEC-RAS) show a good depiction of the impact of the different threshold values. There is a backscatter analysis with the mean sigma db values and the STD for the 9 classes, demonstrating the db values were a class can be found. There is a good visual for the mean db values for each of the 9 classes for all the polarizations. Also, the visualization depicted which classes were miss-classified as wet/dry.

However, there are several weaknesses that should be considered, ordered below from the major to the minor ones.

Major comments

Section 2 has only one subsection 2.2 starting in line 80. The study site should be presented as part of the validation of the proposed method.

In the evaluation part of the method, there is no performance comparison with other thresholding techniques or other similar methods. Please consider some recent works from the literature that propose other ways to select optimal values of the thresholds.

In the way that the paper is structured, it appears that this work is too focused on a specific area of interest and it does not generalize to other areas of interest. The dataset is presented in section 3.1, while it should be reported after the presentation of the methodology for validating and evaluating the methodology.

The methodology is presented in Section 3.2, namely “Hammock Swing Thresholding (HST)”. In this methodology a thresholding approach is applied first on the permanent waters, then on the satellite images showing flood events and the results are compared to provide the flood inundation areas. Please provide an algorithm, showing a list of steps, starting from a well-defined input, in order to show the flow of information towards the generation of the final output.

Other works are properly cited throughout the paper, but please highlight the parts of the present work that are considered novel and not just an incremental contribution.

Section 3.2.5 describes a generation of an image, but there is no reference to a Figure with an example generated result, so please add one.

Minor comments are:

“abs” in Equation (1) could be omitted since the symbol “||” denotes absolute value. Does the dot in Equation (2) denote scalar product? The manuscript should be checked again for typos. For example, in the list of authors, Ngo Duc Anh1 has no superscript in “1”, a full stop appears in line 44, etc. Line 248 math type is missing. Various minor typos are spotted through the manuscript. E.g. line 27: “201”, line 42: “can classify robustly”, line 126: “1849 is actually the Water supply paper number, the year is 1967 as it correctly appears at the references”, line 309: “comparison of this images”, line 383: “deferent times” In addition, the use of English language should be revisited, e.g. “There are many other dams have been projected across the basin.

Author Response

Response to Reviewer 1 Comments

On behalf of the authors, I would like to thank you very much for your constructive comments.

Open Review

When comparing the calibrated value to the default ones, there are times that the variation is big. The “Road” coefficient is changed from 0.032 to 0.78, the “Residence” from 0.068 to 0.760. Are such deviations common?

Response 1: The variations are not common as the Road type and residence in the study area are uncommon too; the most roads are also dykes with the elevation above 4 m (mean elevation is about 1 m) and much vegetation asides. Similarly, residence areas are located in higher ground with dense of houses and vegetation to prevent floods. Base on the field survey the roads and residence were not flooded hence in the calibration phase we increased the road and residence and 3 season rice N to avoid it flooded. Noted that the 3 season rice is barriered by lower dykes (3.5 m) as well. Therefore, the variations were considered reasonable.

Major comments

Section 2 has only one subsection 2.2 starting in line 80. The study site should be presented as part of the validation of the proposed method.

Response 2: This was an editing error. The paper is re-structured and this section is placed in the section 2 as a sub-heading (2.1). We have referred some papers published in the water, each paper structured differently, some papers organized like that we re-structured this manuscript.

In the evaluation part of the method, there is no performance comparison with other thresholding techniques or other similar methods. Please consider some recent works from the literature that propose other ways to select optimal values of the thresholds.

 Response 3: we added a paragraph discussion uncertainty sources and compare to published threshold results in the result section (the second paragraph of the section 3.5). As this HST was first time applied, it is difficult to compare with other methods and the HST was developed basically in inverse way with most exiting ones (normally using remote sensing-based results to validate hydrological models).

In the way that the paper is structured, it appears that this work is too focused on a specific area of interest and it does not generalize to other areas of interest. The dataset is presented in section 3.1, while it should be reported after the presentation of the methodology for validating and evaluating the methodology.

Response 4: The manuscript is re-structured with the data section presented after the methodology section.

The methodology is presented in Section 3.2, namely “Hammock Swing Thresholding (HST)”. In this methodology a thresholding approach is applied first on the permanent waters, then on the satellite images showing flood events and the results are compared to provide the flood inundation areas. Please provide an algorithm, showing a list of steps, starting from a well-defined input, in order to show the flow of information towards the generation of the final output.

 Response 5: we added a graphic HST algorithm with steps to generate final thresholds and short explanations of the algorithm in this section.

Other works are properly cited throughout the paper, but please highlight the parts of the present work that are considered novel and not just an incremental contribution.

Response 6: HST algorithm part using HEC-RAS outputs to estimate optimal water thresholds is considered a novel contribution of this work. It is mentioned in the abstract and in the introduction.

Section 3.2.5 describes a generation of an image, but there is no reference to a Figure with an example generated result, so please add one.

Author response 7: the section Section 3.2.5 is a part of methodology section showing how to achieve the inundation maps. I added to the last sentence and reference to the results section.

 Minor comments are:

 “abs” in Equation (1) could be omitted since the symbol “||” denotes absolute value.

Author response: “abs” have been omitted and also in the equation 4.

Does the dot in Equation (2) denote scalar product?

Author response: the dot in Equation (2) is the multiple operation. Corrected in the revised version.

The manuscript should be checked again for typos. For example, in the list of authors, Ngo Duc Anh1 has no superscript in “1”, a full stop appears in line 44, etc. Line 248 math type is missing. Various minor typos are spotted through the manuscript. E.g. line 27: “201”, line 42: “can classify robustly”, line 126: “1849 is actually the Water supply paper number, the year is 1967 as it correctly appears at the references”, line 309: “comparison of this images”, line 383: “deferent times”.

Author response: thank you for your checks carefully, we have checked typos throughoutly and correct not only errors you pointed to but also the others.

In addition, the use of English language should be revisited, e.g. “There are many other dams have been projected across the basin.

Dr. Christopher R Hackney, a native English speaker, has checked for the use of English language in the manuscript.

Reviewer 2 Report

English form requires extensive revision.

I recommend an overall restructuring of the paper.

Major points:
The whole study relies on the fact that a numerical model can be used to calibrate the wet/dry threshold of the satellite observation. This implies that the employed numerical model chain has uncertainty bounds lower than the satellite data. One could say that, in order to apply the proposed procedure, this constraint has to be fulfilled. My concern is that most of the times this requirement is not met, since the hydrological/hydraulic modelling leads to predictions in terms of flooded areas with uncertainty levels of the order of +- 20/30%. The authors should include this point in the study and provide some discussion along these lines.

Minor points:

Why do you refer to HEC-RAS as a hydrological model? it is a hydraulic model. If you are referring to a hydrologic modelling chain then you should specify the modules employed (HEC-HMS?). Moreover, the sections devoted to the description of the hydrologic/hydraulic modelling should be enlarged, also with reference to the major point above.

Lines 45: the potential reader would benefit from knowing that SAR data can be employed not only to calibrate hydraulic models (as the authors correctly asknowledge (refs 4,9,10) bu also to understand which type of model best suits the underlying dynamics of flooding under consideration see reference [1] below.

line 66. first name of reference [6] is wrong.

line 110 typos

please give a justification for the naming of the method (HST)

The manuscript is full of acronyms, many of which, though known to the experts in the field, are not defined (SNAP, SLC, GRD, RS and many others)

The conclusions seem to refer to a completely different study as they incorporate statements never covered in the manuscript: for example, they mention the need to match resolution of model and SAR data, which is an interesting topic, but is never mentioned in the discussion of the results. Even more surprisingly, the conclusion refer to an "iterative" (line 412) procedure which is not described in the paper....

[1]P. Prestininzi, G. Di Baldassarre, G. Schumann, P.D. Bates, Selecting the appropriate hydraulic model structure using low-resolution satellite imagery,
Advances in Water Resources, Volume 34, Issue 1, 2011, Pages 38-46.

Author Response

Response to Reviewer 2 Comments

On behalf of the authors, I would like to thank you very much for your constructive comments.

Open Review

English form requires extensive revision.

Response 1: Dr. Christopher R Hackney, a native English speaker has checked for the use of English language in the manuscript carefully.

I recommend an overall restructuring of the paper.

Response 2: the paper is restructured referring to the author guides of the journal.

Major points:
3. The whole study relies on the fact that a numerical model can be used to calibrate the wet/dry threshold of the satellite observation. This implies that the employed numerical model chain has uncertainty bounds lower than the satellite data. One could say that, in order to apply the proposed procedure, this constraint has to be fulfilled. My concern is that most of the times this requirement is not met, since the hydrological/hydraulic modelling leads to predictions in terms of flooded areas with uncertainty levels of the order of +- 20/30%. The authors should include this point in the study and provide some discussion along these lines.

Response 3: Thank you for this suggestion, it is an interesting point to discuss. I added this point to the discussion of the flood maps result and discussion. A new paragraph is added to section 3.6. SAR Flood Maps.

Minor points:

Why do you refer to HEC-RAS as a hydrological model? it is a hydraulic model. If you are referring to a hydrologic modelling chain then you should specify the modules employed (HEC-HMS?). Moreover, the sections devoted to the description of the hydrologic/hydraulic modelling should be enlarged, also with reference to the major point above.

Response 4: HEC-RAS is initially known as a hydraulic model which concerning more how water conveyed through pipes, channels and other hydraulic structures. However, in recent years the HEC-RAS model applied more for natural systems on land and rivers. In this study we used just the HEC-RAS 2D which has ability to model large real areas particularly for model flood inundation. Therefore, we refer HEC-RAS 2D as a hydrological model.

Lines 45: the potential reader would benefit from knowing that SAR data can be employed not only to calibrate hydraulic models (as the authors correctly asknowledge (refs 4,9,10) bu also to understand which type of model best suits the underlying dynamics of flooding under consideration see reference [1] below.

Response 5: This is a very good point to discuss a beneficial for readers in both remote sensing and hydrology. I referred this reference in the discussion connect to the major point.

line 66. first name of reference [6] is wrong.

Response 6: Corrected

line 110 typos

Response 7: Corrected

please give a justification for the naming of the method (HST)

Response 8: a justification is added to the thresholding section.

The manuscript is full of acronyms, many of which, though known to the experts in the field, are not defined (SNAP, SLC, GRD, RS and many others)

Response 9: SNAP is the name of the satellite image processing software particularly for Sentinel images provided freely be European Space Agency (ESA) (http://step.esa.int/main/download/snap-download/), hence its definition is not available. Definations of SLC, GRD, RS and some other acronyms are added.

The conclusions seem to refer to a completely different study as they incorporate statements never covered in the manuscript: for example, they mention the need to match resolution of model and SAR data, which is an interesting topic, but is never mentioned in the discussion of the results. Even more surprisingly, the conclusion refer to an "iterative" (line 412) procedure which is not described in the paper....

Response 10:

- The references cited in the conclusion were just for the examples of studies using thresholding method for SAR images. We have not incorporated with statements in those studies. I move the citations to the second line of the conclusion to make the point clearer.

- The "iterative" is described in the method (Figure 1, showing moving Ti = Tini+ interal), All the equations (1) shows Ti =1 to n (n is a number of threshold adjustments (Ti) for the SAR images). In addition, as a requirement of other reviewer, we added a graph presenting steps of the HST algorithm, the iterative is also describes by the “if then” operation (please see the revised version).

[1]P. Prestininzi, G. Di Baldassarre, G. Schumann, P.D. Bates, Selecting the appropriate hydraulic model structure using low-resolution satellite imagery,
Advances in Water Resources, Volume 34, Issue 1, 2011, Pages 38-46.

Round 2

Reviewer 1 Report

The revised version is significantly improved with respect to the original one.

Author Response

Response to Reviewer 1 Comments_V2

On behalf of the authors, I would like to thank you very much again for reviewing our paper.

Open Review

English language and style are fine/minor spell check required

Response 1: We have checked for the language and typos for entire the paper carefully.

Reviewer 2 Report

I'm satisfied with the revision.

I still believe that referring to the hec-ras package as an hydrological model is wrong, and misleading for the hydrological community.

Author Response

Response to Reviewer 2 Comment_V2

On behalf of the authors, I would like to thank you very much again for reviewing our paper.

Open Review

I still believe that referring to the hec-ras package as an hydrological model is wrong, and misleading for the hydrological community.

Response 1: We have discussed this point again and agreed with you to change this term of hydrological model to "hydraulic model" when it is specificly pointed to the HEC-RAS model.

In the general context of using hydrological/hydraulic model results for the HST method, we open for both uses of both the model types. Therefore, we changed the word “hydrological” to “hydrological/hydraulic” in this context. Changes are marked in red in the revised manuscript.

Thank you very much one again!
